# Fully Decentralized Policies for Multi-Agent Systems: An Information Theoretic Approach

**Roel Dobbe**[*]
Electrical Engineering and Computer Science
University of California, Berkeley
Berkeley, CA 94720
dobbe@eecs.berkeley.edu

**David Fridovich-Keil**[*]
Electrical Engineering and Computer Science
University of California, Berkeley
Berkeley, CA 94720
dfk@eecs.berkeley.edu

**Claire Tomlin**
Electrical Engineering and Computer Science
University of California, Berkeley
Berkeley, CA 94720
tomlin@eecs.berkeley.edu

## Abstract

Learning cooperative policies for multi-agent systems is often challenged by partial observability and a lack of coordination. In some settings, the structure of a problem allows a distributed solution with limited communication. Here, we consider a scenario where no communication is available, and instead we learn local policies for all agents that collectively mimic the solution to a centralized multi-agent static optimization problem. Our main contribution is an information theoretic framework based on rate distortion theory which facilitates analysis of how well the resulting fully decentralized policies are able to reconstruct the optimal solution. Moreover, this framework provides a natural extension that addresses which nodes an agent should communicate with to improve the performance of its individual policy.

## 1 Introduction

Finding optimal decentralized policies for multiple agents is often a hard problem hampered by partial observability and a lack of coordination between agents. The distributed multi-agent problem has been approached from a variety of angles, including distributed optimization [Boyd et al., 2011], game theory [Aumann and Dreze, 1974] and decentralized or networked partially observable Markov decision processes (POMDPs) [Oliehoek and Amato, 2016, Goldman and Zilberstein, 2004, Nair et al., 2005]. In this paper, we analyze a different approach consisting of a simple learning scheme to design fully decentralized policies for all agents that collectively mimic the solution to a common optimization problem, while having no access to a global reward signal and either no or restricted access to other agents' local state. This algorithm is a generalization of that proposed in our prior work [Sondermeijer et al., 2016] related to decentralized optimal power flow (OPF). Indeed, the success of regression-based decentralization in the OPF domain motivated us to understand when and how well the method works in a more general decentralized optimal control setting.

The key contribution of this work is to view decentralization as a *compression* problem, and then apply classical results from information theory to analyze performance limits. More specifically, we treat the $i^{\text{th}}$ agent's optimal action in the centralized problem as a random variable $u_i^*$, and model its conditional dependence on the global state variables $x = (x_1, \ldots, x_n)$, i.e. $p(u_i^* | x)$, which we

---
[*]Indicates equal contribution.

assume to be stationary in time. We now restrict each agent $i$ to observe only the $i^{\text{th}}$ state variable $x_i$. Rather than solving this decentralized problem directly, we train each agent to replicate what it would have done with full information in the centralized case. That is, the vector of state variables $x$ is *compressed*, and the $i^{\text{th}}$ agent must decompress $x_i$ to compute some estimate $\hat{u}_i \approx u_i^*$. In our approach, each agent learns a parameterized Markov control policy $\hat{u}_i = \hat{\pi}_i(x_i)$ via regression. The $\hat{\pi}_i$ are learned from a data set containing local states $x_i$ taken from historical measurements of system state $x$ and corresponding optimal actions $u_i^*$ computed by solving an offline centralized optimization problem for each $x$.

In this context, we analyze the fundamental limits of compression. In particular, we are interested in unraveling the relationship between the dependence structure of $u_i^*$ and $x$ and the corresponding ability of an agent with partial information to approximate the optimal solution, i.e. the difference – or *distortion* – between decentralized action $\hat{u}_i = \hat{\pi}_i(x_i)$ and $u_i^*$. This type of relationship is well studied within the information theory literature as an instance of *rate distortion theory* [Cover and Thomas, 2012, Chapter 13]. Classical results in this field provide a means of finding a lower bound on the expected distortion as a function of the mutual information – or *rate* of communication – between $u_i^*$ and $x_i$. This lower bound is valid for each specified distortion metric, and for *any* arbitrary strategy of computing $\hat{u}_i$ from available data $x_i$. Moreover, we are able to leverage a similar result to provide a conceptually simple algorithm for choosing a communication structure – letting the regressor $\hat{\pi}_i$ depend on some other local states $x_{j \neq i}$ – in such a way that the lower bound on expected distortion is minimized. As such, our method generalizes [Sondermeijer et al., 2016] and provides a novel approach for the design and analysis of regression-based decentralized optimal policies for general multi-agent systems. We demonstrate these results on synthetic examples, and on a real example drawn from solving OPF in electrical distribution grids.

## 2 Related Work

Decentralized control has long been studied within the system theory literature, e.g. [Lunze, 1992, Siljak, 2011]. Recently, various decomposition based techniques have been proposed for distributed optimization based on primal or dual decomposition methods, which all require iterative computation and some form of communication with either a central node [Boyd et al., 2011] or neighbor-to-neighbor on a connected graph [Pu et al., 2014, Raffard et al., 2004, Sun et al., 2013]. Distributed model predictive control (MPC) optimizes a networked system composed of subsystems over a time horizon, which can be decentralized (no communication) if the dynamic interconnections between subsystems are weak in order to achieve closed-loop stability as well as performance [Christofides et al., 2013]. The work of Zeilinger et al. [2013] extended this to systems with strong coupling by employing time-varying distributed terminal set constraints, which requires neighbor-to-neighbor communication. Another class of methods model problems in which agents try to cooperate on a common objective without full state information as a decentralized partially observable Markov decision process (Dec-POMDP) [Oliehoek and Amato, 2016]. Nair et al. [2005] introduce networked distributed POMDPs, a variant of the Dec-POMDP inspired in part by the pairwise interaction paradigm of distributed constraint optimization problems (DCOPs).

Although the specific algorithms in these works differ significantly from the regression-based decentralization scheme we consider in this paper, a larger difference is in problem formulation. As described in Sec. 3, we study a static optimization problem repeatedly solved at each time step. Much prior work, especially in optimal control (e.g. MPC) and reinforcement learning (e.g. Dec-POMDPs), poses the problem in a dynamic setting where the goal is to minimize cost over some time horizon. In the context of reinforcement learning (RL), the time horizon can be very long, leading to the well known tradeoff between exploration and exploitation; this does not appear in the static case. Additionally, many existing methods for the dynamic setting require an ongoing communication strategy between agents – though not all, e.g. [Peshkin et al., 2000]. Even one-shot static problems such as DCOPs tend to require complex communication strategies, e.g. [Modi et al., 2005].

Although the mathematical formulation of our approach is rather different from prior work, the policies we compute are similar in spirit to other learning and robotic techniques that have been proposed, such as behavioral cloning [Sammut, 1996] and apprenticeship learning [Abbeel and Ng, 2004], which aim to let an agent learn from examples. In addition, we see a parallel with recent work on information-theoretic bounded rationality [Ortega et al., 2015] which seeks to formalize decision-making with limited resources such as the time, energy, memory, and computational effort

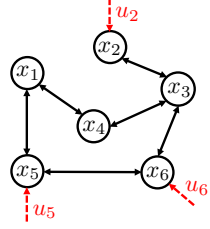
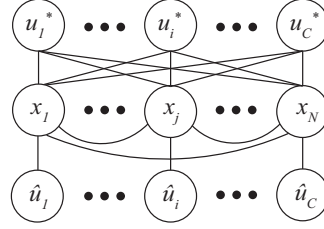

(a) Distributed multi-agent problem.               (b) Graphical model of dependency structure.

Figure 1: (a) shows a connected graph corresponding to a distributed multi-agent system. The circles denote the local state $x_i$ of an agent, the dashed arrow denotes its action $u_i$, and the double arrows denote the physical coupling between local state variables. (b) shows the Markov Random Field (MRF) graphical model of the dependency structure of all variables in the decentralized learning problem. Note that the state variables $x_i$ and the optimal actions $u_i^*$ form a fully connected undirected network, and the local policy $\hat{u}_i$ only depends on the local state $x_i$.

allocated for arriving at a decision. Our work is also related to swarm robotics [Brambilla et al., 2013], as it learns simple rules aimed to design robust, scalable and flexible collective behaviors for coordinating a large number of agents or robots.

## 3 General Problem Formulation

Consider a distributed multi-agent problem defined by a graph $\mathcal{G} = (\mathcal{N}, \mathcal{E})$, with $\mathcal{N}$ denoting the nodes in the network with cardinality $|\mathcal{N}| = N$, and $\mathcal{E}$ representing the set of edges between nodes. Fig. 1a shows a prototypical graph of this sort. Each node has a real-valued state vector $x_i \in \mathbb{R}^{\alpha_i}, i \in \mathcal{N}$. A subset of nodes $\mathcal{C} \subset \mathcal{N}$, with cardinality $|\mathcal{C}| = C$, are controllable and hence are termed "agents." Each of these agents has an action variable $u_i \in \mathbb{R}^{\beta_i}, i \in \mathcal{C}$. Let $x = (x_i, \ldots, x_N)^\top \in \mathbb{R}^{\sum_{i \in \mathcal{N}} \alpha_i} = \mathcal{X}$ denote the full network state vector and $u \in \mathbb{R}^{\sum_{i \in \mathcal{C}} \beta_i} = \mathcal{U}$ the stacked network optimization variable. Physical constraints such as spatial coupling are captured through equality constraints $g(x, u) = 0$. In addition, the system is subject to inequality constraints $h(x, u) \leq 0$ that incorporate limits due to capacity, safety, robustness, etc. We are interested in minimizing a convex scalar function $f_o(x, u)$ that encodes objectives that are to be pursued cooperatively by all agents in the network, i.e. we want to find

$$
\begin{aligned}
u^* = \arg\min_u \quad & f_o(x, u), \\
\text{s.t.} \quad & g(x, u) = 0, \quad h(x, u) \leq 0.
\end{aligned}
\tag{1}
$$

Note that (1) is static in the sense that it does not consider the future evolution of the state $x$ or the corresponding future values of cost $f_o$. We apply this static problem to sequential control tasks by repeatedly solving (1) at each time step. Note that this simplification from an explicitly dynamic problem formulation (i.e. one in which the objective function incorporates future costs) is purely for ease of exposition and for consistency with the OPF literature as in [Sondermeijer et al., 2016]. We could also consider the optimal policy which solves a dynamic optimal control or RL problem and the decentralized learning step in Sec. 3.1 would remain the same.

Since (1) is static, applying the learned decentralized policies repeatedly over time may lead to dynamical instability. Identifying when this will and will not occur is a key challenge in verifying the regression-based decentralization method, however it is beyond the scope of this work.

### 3.1 Decentralized Learning

We interpret the process of solving (1) as applying a well-defined function or stationary Markov policy $\pi^* : \mathcal{X} \longrightarrow \mathcal{U}$ that maps an input collective state $x$ to the optimal collective control or action $u^*$. We presume that this solution exists and can be computed offline. Our objective is to learn $C$ decentralized policies $\hat{u}_i = \hat{\pi}_i(x_i)$, one for each agent $i \in \mathcal{C}$, based on $T$ historical measurements of the states $\{x[t]\}_{t=1}^T$ and the offline computation of the corresponding optimal actions $\{u^*[t]\}_{t=1}^T$. Although each policy $\hat{\pi}_i$ individually aims to approximate $u_i^*$ based on local state $x_i$, we are able

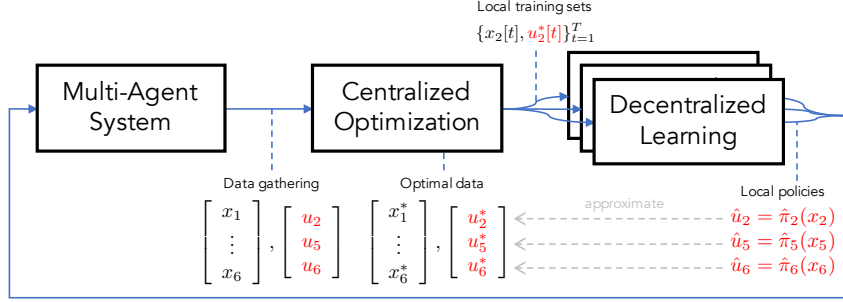

Figure 2: A flow diagram explaining the key steps of the decentralized regression method, depicted for the example system in Fig. 1a. We first collect data from a multi-agent system, and then solve the centralized optimization problem using all the data. The data is then split into smaller training and test sets for all agents to develop individual decentralized policies $\hat{\pi}_i(x_i)$ that approximate the optimal solution of the centralized problem. These policies are then implemented in the multi-agent system to collectively achieve a common global behavior.

to reason about how well their collective action can approximate $\pi^*$. Figure 2 summarizes the decentralized learning setup.

More formally, we describe the dependency structure of the individual policies $\hat{\pi}_i : \mathbb{R}^{\alpha_i} \longrightarrow \mathbb{R}^{\beta_i}$ with a Markov Random Field (MRF) graphical model, as shown in Fig. 1b. The $\hat{u}_i$ are only allowed to depend on local state $x_i$ while the $u_i^*$ may depend on the full state $x$. With this model, we can determine how information is distributed among different variables and what information-theoretic constraints the policies $\{\hat{\pi}_i\}_{i \in \mathcal{C}}$ are subject to when collectively trying to reconstruct the centralized policy $\pi^*$. Note that although we may refer to $\pi^*$ as globally optimal, this is not actually required for us to reason about how closely the $\hat{\pi}_i$ approximate $\pi^*$. That is, our analysis holds even if (1) is solved using approximate methods. In a dynamical reformulation of (1), for example, $\pi^*$ could be generated using techniques from deep RL.

### 3.2 A Rate-Distortion Framework

We approach the problem of how well the decentralized policies $\hat{\pi}_i$ can perform in theory from the perspective of *rate distortion*. Rate distortion theory is a sub-field of information theory which provides a framework for understanding and computing the minimal *distortion* incurred by any given *compression* scheme. In a rate distortion context, we can interpret the fact that the output of each individual policy $\hat{\pi}_i$ depends only on the local state $x_i$ as a compression of the full state $x$. For a detailed overview, see [Cover and Thomas, 2012, Chapter 10]. We formulate the following variant of the the classical rate distortion problem

$$D^* = \min_{p(\hat{u}|u^*)} \quad \mathbb{E}\left[d(\hat{u}, u^*)\right] , \tag{2}$$
$$\text{s.t.} \quad I(\hat{u}_i; u_j^*) \leq I(x_i; u_j^*) \triangleq \gamma_{ij} ,$$
$$I(\hat{u}_i; \hat{u}_j) \leq I(x_i; x_j) \triangleq \delta_{ij}, \forall i, j \in \mathcal{C} ,$$

where $I(\cdot, \cdot)$ denotes mutual information and $d(\cdot, \cdot)$ an arbitrary non-negative distortion measure. As usual, the minimum distortion between random variable $u^*$ and its reconstruction $\hat{u}$ may be found by minimizing over conditional distributions $p(\hat{u}|u^*)$.

The novelty in (2) lies in the structure of the constraints. Typically, $D^*$ is written as a function $D(R)$, where $R$ is the maximum *rate* or mutual information $I(\hat{u}; u^*)$. From Fig. 1b however, we know that pairs of reconstructed and optimal actions cannot share more information than is contained in the intermediate nodes in the graphical model, e.g. $\hat{u}_1$ and $u_1^*$ cannot share more information than $x_1$ and $u_1^*$. This is a simple consequence of the data processing inequality [Cover and Thomas, 2012, Thm. 2.8.1]. Similarly, the reconstructed optimal actions at two different nodes cannot be more closely related than the measurements $x_i$'s from which they are computed. The resulting constraints are fixed by the joint distribution of the state $x$ and the optimal actions $u^*$. That is, they are fully determined by the structure of the optimization problem (1) that we wish to solve.

We emphasize that we have made virtually no assumptions about the distortion function. For the remainder of this paper, we will measure distortion as the deviation between $\hat{u}_i$ and $u_i^*$. However, we could also define it to be the suboptimality gap $f_o(x, \hat{u}) - f_o(x, u^*)$, which may be much more complicated to compute. This definition could allow us to reason explicitly about the cost of decentralization, and it could address the valid concern that the optimal decentralized policy may bear no resemblance to $\pi^*$. We leave further investigation for future work.

### 3.3 Example: Squared Error, Jointly Gaussian

To provide more intuition into the rate distortion framework, we consider an idealized example in which the $x_i, u_i \in \mathbb{R}^1$. Let $d(\hat{u}, u^*) = \|\hat{u} - u^*\|_2^2$ be the squared error distortion measure, and assume the state $x$ and optimal actions $u^*$ to be jointly Gaussian. These assumptions allow us to derive an explicit formula for the optimal distortion $D^*$ and corresponding regression policies $\hat{\pi}_i$. We begin by stating an identity for two jointly Gaussian $X, Y \in \mathbb{R}$ with correlation $\rho$: $I(X; Y) \leq \gamma \iff \rho^2 \leq 1 - e^{-2\gamma}$, which follows immediately from the definition of mutual information and the formula for the entropy of a Gaussian random variable. Taking $\rho_{\hat{u}_i, u_i^*}$ to be the correlation between $\hat{u}_i$ and $u_i^*$, $\sigma_{\hat{u}_i}^2$ and $\sigma_{u_i^*}^2$ to be the variances of $\hat{u}_i$ and $u_i^*$ respectively, and assuming that $u_i^*$ and $\hat{u}_i$ are of equal mean (unbiased policies $\hat{\pi}_i$), we can show that the minimum distortion attainable is

$$D^* = \min_{p(\hat{u}|u^*)} \mathbb{E}\left[\|u^* - \hat{u}\|_2^2\right] : \rho_{\hat{u}_i, u_i^*}^2 \leq 1 - e^{-2\gamma_{ii}} = \rho_{u_i^*, x_i}^2, \forall i \in \mathcal{C}, \tag{3}$$

$$= \min_{\{\rho_{\hat{u}_i, u_i^*}\}, \{\sigma_{\hat{u}_i}\}} \sum_i \left(\sigma_{u_i^*}^2 + \sigma_{\hat{u}_i}^2 - 2\rho_{\hat{u}_i, u_i^*}\sigma_{u_i^*}\sigma_{\hat{u}_i}\right) : \rho_{\hat{u}_i, u_i^*}^2 \leq \rho_{u_i^*, x_i}^2, \tag{4}$$

$$= \min_{\{\sigma_{\hat{u}_i}\}} \sum_i \left(\sigma_{u_i^*}^2 + \sigma_{\hat{u}_i}^2 - 2\rho_{u_i^*, x_i}\sigma_{u_i^*}\sigma_{\hat{u}_i}\right), \tag{5}$$

$$= \sum_i \sigma_{u_i^*}^2 (1 - \rho_{u_i^*, x_i}^2). \tag{6}$$

In (4), we have solved for the optimal correlations $\rho_{\hat{u}_i, u_i^*}$. Unsurprisingly, the optimal value turns out to be the maximum allowed by the mutual information constraint, i.e. $\hat{u}_i$ should be as correlated to $u_i^*$ as possible, and in particular as much as $u_i^*$ is correlated to $x_i$. Similarly, in (5) we solve for the optimal $\sigma_{\hat{u}_i}$, with the result that at optimum, $\sigma_{\hat{u}_i} = \rho_{u_i^*, x_i}\sigma_{u_i^*}$. This means that as the correlation between the local state $x_i$ and the optimal action $u_i^*$ decreases, the variance of the estimated action $\hat{u}_i$ decreases as well. As a result, the learned policy will increasingly "bet on the mean" or "listen less" to its local measurement to approximate the optimal action.

Moreover, we may also provide a closed form expression for the regressor which achieves the minimum distortion $D^*$. Since we have assumed that each $u_i^*$ and the state $x$ are jointly Gaussian, we may write any $u_i^*$ as an affine function of $x_i$ plus independent Gaussian noise. Thus, the minimum mean squared estimator is given by the conditional expectation

$$\hat{u}_i = \hat{\pi}_i(x_i) = \mathbb{E}\left[u_i^* | x_i\right] = \mathbb{E}\left[u_i^*\right] + \frac{\rho_{u_i^* x_i}\sigma_{u_i^*}}{\sigma_{x_i}}(x_i - \mathbb{E}\left[x_i\right]). \tag{7}$$

Thus, we have found a closed form expression for the best regressor $\hat{\pi}_i$ to predict $u_i^*$ from only $x_i$ in the joint Gaussian case with squared error distortion. This result comes as a direct consequence of knowing the true parameterization of the joint distribution $p(u^*, x)$ (in this case, as a Gaussian).

### 3.4 Determining Minimum Distortion in Practice

Often in practice, we do not know the parameterization $p(u^*|x)$ and hence it may be intractable to determine $D^*$ and the corresponding decentralized policies $\hat{\pi}_i$. However, if one can assume that $p(u^*|x)$ belongs to a family of parameterized functions (for instance universal function approximators such as deep neural networks), then it is theoretically possible to attain or at least approach minimum distortion for arbitrary non-negative distortion measures.

Practically, one can compute the mutual information constraint $I(u_i^*, x_i)$ from (2) to understand how much information a regressor $\hat{\pi}_i(x_i)$ has available to reconstruct $u_i^*$. In the Gaussian case, we were able to compute this mutual information in closed form. For data from general distributions

however, there is often no way to compute mutual information analytically. Instead, we rely on access to sufficient data $\{x[t], u^*[t]\}_{t=1}^T$, in order to estimate mutual informations numerically. In such situations (e.g. Sec. 5), we discretize the data and then compute mutual information with a minimax risk estimator, as proposed by Jiao et al. [2014].

# 4 Allowing Restricted Communication

Suppose that a decentralized policy $\hat{\pi}_i$ suffers from insufficient mutual information between its local measurement $x_i$ and the optimal action $u_i^*$. In this case, we would like to quantify the potential benefits of communicating with other nodes $j \neq i$ in order to reduce the distortion limit $D^*$ from (2) and improve its ability to reconstruct $u_i^*$. In this section, we present an information-theoretic solution to the problem of how to choose optimally which other data to observe, and we provide a lower bound-achieving solution for the idealized Gaussian case introduced in Sec. 3.3. We assume that in addition to observing its own local state $x_i$, each $\hat{\pi}_i$ is allowed to depend on at most $k$ other $x_{j \neq i}$.

**Theorem 1.** *(Restricted Communication)*

*If $\mathcal{S}_i$ is the set of $k$ nodes $j \neq i \in \mathcal{N}$ which $\hat{u}_i$ is allowed to observe in addition to $x_i$, then setting*

$$\mathcal{S}_i = \arg\max_{\mathcal{S}} \ I(u_i^*; x_i, \{x_j : j \in \mathcal{S}\}) \ : \ |\mathcal{S}| = k \,, \tag{8}$$

*minimizes the best-case expectation of any distortion measure. That is, this choice of $\mathcal{S}_i$ yields the smallest lower bound $D^*$ from (2) of any possible choice of $\mathcal{S}$.*

*Proof.* By assumption, $\mathcal{S}_i$ maximizes the mutual information between the observed local states $\{x_i, \ x_j \ : \ j \in \mathcal{S}_i\}$ and the optimal action $u_i^*$. This mutual information is equivalent to the notion of *rate $R$* in the classical rate distortion theorem [Cover and Thomas, 2012]. It is well-known that the distortion rate function $D(R)$ is convex and monotone decreasing in $R$. Thus, by maximizing mutual information $R$ we are guaranteed to minimize distortion $D(R)$, and hence $D^*$. $\qquad\square$

Theorem 1 provides a means of choosing a subset of the state $\{x_j : j \neq i\}$ to communicate to each decentralized policy $\hat{\pi}_i$ that minimizes the corresponding best expected distortion $D^*$. Practically speaking, this result may be interpreted as formalizing the following intuition: "the best thing to do is to transmit the most information." In this case, "transmitting the most information" corresponds to allowing $\hat{\pi}_i$ to observe the set $\mathcal{S}$ of nodes $\{x_j \ : \ j \neq i\}$ which contains the most information about $u_i^*$. Likewise, by "best" we mean that $\mathcal{S}_i$ minimizes the best-case expected distortion $D^*$, for any distortion metric $d$. As in Sec. 3.3, without making some assumption about the structure of the distribution of $x$ and $u^*$, we cannot guarantee that any particular regressor $\hat{\pi}_i$ will attain $D^*$. Nevertheless, in a practical situation where sufficient data $\{x[t], u^*[t]\}_{t=1}^T$ is available, we can solve (8) by estimating mutual information [Jiao et al., 2014].

## 4.1 Example: Joint Gaussian, Squared Error with Communication

Here, we reexamine the joint Gaussian-distributed, mean squared error distortion case from Sec. 3.3, and apply Thm. 1. We will take $u^* \in \mathbb{R}^1, x \in \mathbb{R}^{10}$ and $u^*, x$ jointly Gaussian with zero mean and arbitrary covariance. The specific covariance matrix $\Sigma$ of the joint distribution $p(u^*, x)$ is visualized in Fig. 3a. For simplicity, we show the squared correlation coefficients of $\Sigma$ which lie in $[0, 1]$. The boxed cells in $\Sigma$ in Fig. 3a indicate that $x_9$ solves (8), i.e. $j = 9$ maximizes $I(u^*; x_1, x_j)$ the mutual information between the observed data and regression target $u^*$. Intuitively, this choice of $j$ is best because $x_9$ is highly correlated to $u^*$ and weakly correlated to $x_1$, which is already observed by $\hat{u}$; that is, it conveys a significant amount of information about $u^*$ that is not already conveyed by $x_1$.

Figure 3b shows empirical results. Along the horizontal axis we increase the value of $k$, the number of additional variables $x_j$ which regressor $\hat{\pi}_i$ observes. The vertical axis shows the resulting average distortion. We show results for a linear regressor of the form of (7) where we have chosen $\mathcal{S}_i$ optimally according to (8), as well as uniformly at random from all possible sets of unique indices. Note that the optimal choice of $\mathcal{S}_i$ yields the lowest average distortion $D^*$ for all choices of $k$. Moreover, the linear regressor of (7) achieves $D^*$ for all $k$, since we have assumed a Gaussian joint distribution.

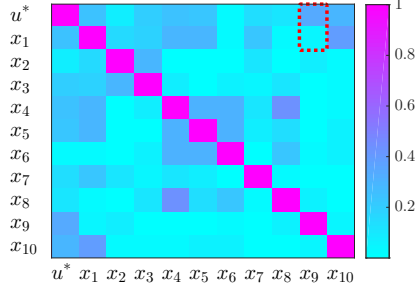
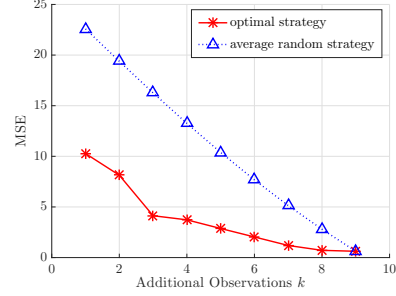

(a) Squared correlation coefficients.　　　　(b) Comparison of communication strategies.

Figure 3: Results for optimal communication strategies on a synthetic Gaussian example. (a) shows squared correlation coefficients between of $u^*$ and all $x_i$'s. The boxed entries correspond to $x_9$, which was found to be optimal for $k = 1$. (b) shows that the optimal communication strategy of Thm. 1 achieves the lowest average distortion and outperforms the average over random strategies.

# 5　Application to Optimal Power Flow

In this case study, we aim to minimize the voltage variability in an electric grid caused by intermittent renewable energy sources and the increasing load caused by electric vehicle charging. We do so by controlling the reactive power output of distributed energy resources (DERs), while adhering to the physics of power flow and constraints due to energy capacity and safety. Recently, various approaches have been proposed, such as [Farivar et al., 2013] or [Zhang et al., 2014]. In these methods, DERs tend to rely on an extensive communication infrastructure, either with a central master node [Xu et al., 2017] or between agents leveraging local computation [Dall'Anese et al., 2014]. We study regression-based decentralization as outlined in Sec. 3 and Fig. 2 to the optimal power flow (OPF) problem [Low, 2014], as initially proposed by Sondermeijer et al. [2016]. We apply Thm. 1 to determine the communication strategy that minimizes optimal distortion to further improve the reconstruction of the optimal actions $u_i^*$.

Solving OPF requires a model of the electricity grid describing both topology and impedances; this is represented as a graph $\mathcal{G} = (\mathcal{N}, \mathcal{E})$. For clarity of exposition and without loss of generality, we introduce the linearized power flow equations over radial networks, also known as the *LinDistFlow* equations [Baran and Wu, 1989]:

$$P_{ij} = \sum_{(j,k)\in\mathcal{E}, k\neq i} P_{jk} + p_j^{\mathrm{c}} - p_j^{\mathrm{g}}, \tag{9a}$$

$$Q_{ij} = \sum_{(j,k)\in\mathcal{E}, k\neq i} Q_{jk} + q_j^{\mathrm{c}} - q_j^{\mathrm{g}}, \tag{9b}$$

$$v_j = v_i - 2\left(r_{ij}P_{ij} + \xi_{ij}Q_{ij}\right) \tag{9c}$$

In this model, capitals $P_{ij}$ and $Q_{ij}$ represent real and reactive power flow on a branch from node $i$ to node $j$ for all branches $(i,j) \in \mathcal{E}$, lower case $p_i^c$ and $q_i^c$ are the real and reactive power consumption at node $i$, and $p_i^g$ and $q_i^g$ are its real and reactive power generation. Complex line impedances $r_{ij} + \sqrt{-1}\xi_{ij}$ have the same indexing as the power flows. The *LinDistFlow* equations use the squared voltage magnitude $v_i$, defined and indexed at all nodes $i \in \mathcal{N}$. These equations are included as constraints in the optimization problem to enforce that the solution adheres to laws of physics.

To formulate our decentralized learning problem, we will treat $x_i \triangleq (p_i^c, q_i^c, p_i^g)$ to be the local state variable, and, for all controllable nodes, i.e. agents $i \in \mathcal{C}$, we have $u_i \triangleq q_i^g$, i.e. the reactive power generation can be controlled ($v_i, P_{ij}, Q_{ij}$ are treated as dummy variables). We assume that for all nodes $i \in \mathcal{N}$, consumption $p_i^c$, $q_i^c$ and real power generation $p_i^g$ are predetermined respectively by the demand and the power generated by a potential photovoltaic (PV) system. The action space is constrained by the reactive power capacity $|u_i| = \left|q_i^{\mathrm{g}}\right| \le \bar{q}_i$. In addition, voltages are maintained

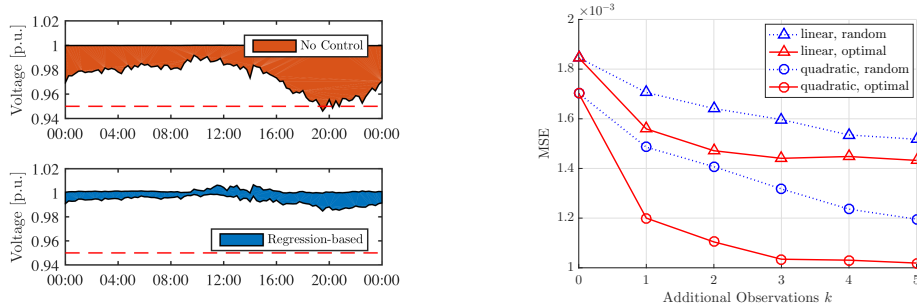

(a) Voltage output with and without control.    (b) Comparison of OPF communication strategies.

Figure 4: Results for decentralized learning on an OPF problem. (a) shows an example result of decentralized learning - the shaded region represents the range of all voltages in a network over a full day. As compared to no control, the fully decentralized regression-based control reduces voltage variation and prevents constraint violation (dashed line). (b) shows that the optimal communication strategy $\mathcal{S}_i$ outperforms the average for random strategies on the mean squared error distortion metric. The regressors used are stepwise linear policies $\hat{\pi}_i$ with linear or quadratic features.

within $\pm 5\%$ of $120V$, which is expressed as the constraint $\underline{v} \leq v_i \leq \overline{v}$. The OPF problem now reads

$$u^* = \arg \min_{q_i^g, \, \forall i \in \mathcal{C}} \quad \sum_{i \in \mathcal{N}} |v_i - v_{\text{ref}}|, \tag{10}$$

$$\text{s.t.} \quad (9), \, \left| q_i^g \right| \leq \bar{q}_i, \, \underline{v} \leq v_i \leq \overline{v}.$$

Following Fig. 2, we employ models of real electrical distribution grids (including the IEEE Test Feeders [IEEE PES, 2017]), which we equip with with $T$ historical readings $\{x[t]\}_{t=1}^T$ of load and PV data, which is composed with real smart meter measurements sourced from Pecan Street Inc. [2017]. We solve (10) for all data, yielding a set of minimizers $\{u^*[t]\}_{t=1}^T$. We then separate the overall data set into $C$ smaller data sets $\{x_i[t], u_i^*[t]\}_{t=1}^T, \, \forall i \in \mathcal{C}$ and train linear policies with feature kernels $\phi_i(\cdot)$ and parameters $\theta_i$ of the form $\hat{\pi}_i(x_i) = \theta_i^\top \phi_i(x_i)$. Practically, the challenge is to select the best feature kernel $\phi_i(\cdot)$. We extend earlier work which showed that decentralized learning for OPF can be done satisfactorily via a hybrid forward- and backward-stepwise selection algorithm [Friedman et al., 2001, Chapter 3] that uses a quadratic feature kernels.

Figure 4a shows the result for an electric distribution grid model based on a real network from Arizona. This network has 129 nodes and, in simulation, 53 nodes were equipped with a controllable DER (i.e. $N = 129, C = 53$). In Fig. 4a we show the voltage deviation from a normalized setpoint on a simulated network with data not used during training. The improvement over the no-control baseline is striking, and performance is nearly identical to the optimum achieved by the centralized solution. Concretely, we observed: (i) no constraint violations, and (ii) a suboptimality deviation of 0.15% on average, with a maximum deviation of 1.6%, as compared to the optimal policy $\pi^*$.

In addition, we applied Thm. 1 to the OPF problem for a smaller network [IEEE PES, 2017], in order to determine the optimal communication strategy to minimize a squared error distortion measure. Fig. 4b shows the mean squared error distortion measure for an increasing number of observed nodes $k$ and shows how the optimal strategy outperforms an average over random strategies.

## 6  Conclusions and Future Work

This paper generalizes the approach of Sondermeijer et al. [2016] to solve multi-agent static optimal control problems with decentralized policies that are learned offline from historical data. Our rate distortion framework facilitates a principled analysis of the performance of such decentralized policies and the design of optimal communication strategies to improve individual policies. These techniques work well on a model of a sophisticated real-world OPF example.

There are still many open questions about regression-based decentralization. It is well known that strong interactions between different subsystems may lead to instability and suboptimality in decentralized control problems [Davison and Chang, 1990]. There are natural extensions of our work

to address dynamic control problems more explicitly, and stability analysis is a topic of ongoing work. Also, analysis of the suboptimality of regression-based decentralization should be possible within our rate distortion framework. Finally, it is worth investigating the use of deep neural networks to parameterize both the distribution $p(u^*|x)$ and local policies $\hat{\pi}_i$ in more complicated decentralized control problems with arbitrary distortion measures.

### Acknowledgments

The authors would like to acknowledge Roberto Calandra for his insightful suggestions and feedback on the manuscript. This research is supported by NSF under the CPS Frontiers VehiCal project (1545126), by the UC-Philippine-California Advanced Research Institute under projects IIID-2016-005 and IIID-2015-10, and by the ONR MURI Embedded Humans (N00014-16-1-2206). David Fridovich-Keil was also supported by the NSF GRFP.

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
