[Reviews · NeurIPS 2017]

Reviewer 1



This paper proposes the use of rate distortion theory to develop (i) decentralized policies and (ii) distributed policies with restricted number of communication for approximating an optimal centralized controller. The problem is very relevant in light of more complex and larger decision problems today, and I find the general idea very interesting. However, some key aspects did not become clear to me and need to be addressed; in particular: is the considered decision problem static or dynamic?; missing discussion on stability; how to solve the centralized problem?; and a discussion on related work on decentralized control. Comments: 1) From the general problem formulation, it did not become clear to me whether the authors consider a static or dynamic decision problem, i.e. is the optimization problem solved once, or is a sequential decision problem considered over a time horizon. The problem (1) is a static problem (u is introduced as the collection of the inputs of all agents, but not over several time steps). However, later the authors typically talk about policies, which alludes to some decision making problem over time. This must be clarified in order to appreciate the result and compare it to others, e.g. those mentioned in related work. I suggest to possibly revisit the problem formulation, and in any case state whether a static or dynamic problem is considered. Also for the example in Section 5, it is not clear if this problem needs to be treated as a dynamic problem, or whether the decision is simply considered for each time individually (and thus can be seen as a static problem). 2) In the introduction, the authors mention relevant work on distributed systems from optimization, game theory, and POMDPs. There is also a large body of work on distributed and decentralized methods from systems and control theory, which should be mentioned in this context (see e.g. book by Siljak “Decentralized control of complex systems,” or Lunze “Feedback control of large-scale systems” on some classical work). Furthermore, in the discussion of Related work in Section 2, the authors point out as a main novelty that they seek decentralized policies with no communication (rather than distributed with some communication). The problem of decentralized control is a classic one in control theory (see e.g. above books), so considering multi-agent policies with no communication is by itself not new at all. The authors should mention this work and balance their statements on novelty. 3) The centralized problem (1) may by itself represent a difficult optimization problem. However, the authors do assume availability of optimal control data {u^*,x} in their later approximations. I think that the authors should discuss, how they expect the centralized problem can be solved and/or clearly state the availability of a solution as an assumption. 4) Unless the authors limit their attention of this work to static decision problems, the aspect of dynamic stability should be mentioned (even if not a contribution of this work). In particular, I would expect that stability can potentially be lost when moving from the centralized policy to the decentralized one because there seems to be no particular objective in the decentralized learning/approximation that would ensure or incentivize a stable controller. This should be discussed in order to make the reader aware of this (possibly critical) aspect of the controller. 5) The authors present the main technical results for a centralized controller whose state and input distribution is jointly Gaussian (sec. 3.3 + 4.1). Is this a reasonable assumption? What decision problems are of this kind (maybe linear problem with Gaussian uncertainty??)? 6) In Section 5, why is the restriction to linear power flow equations without loss of generality? 7) I do not understand figure 4(a). What is the grey versus the white region? What quantity do the graphs represent?

Reviewer 2



In the context of learning cooperative policies for multi-agent systems, it often requires some sort of communications among agents. This paper considers the case where no communication is available. The authors provided an information theoretical framework based on rate distortion theory to solve multi-agent system problem in a near-optimal way with decentralized policies that are learned with data from historical measurements. The authors also provided a natural extension to address which nodes a node should communication with to improve the performance of its individual policy. The proposed framework was theoretically proven and also illustrated by solving an OPT problem in power system operation. The paper is very well written and the results are technically correct. The problem is also huge interesting to the community. Thus, I strong recommend its acceptance. I only has some minor comments regarding the application about DER coordination: There are other recent works which considers DER optimal actions. For example, 1) https://arxiv.org/abs/1610.06631 2) http://ieeexplore.ieee.org/document/7961194/ If possible, could you please compare your work in this regard with the existing ones?

Reviewer 3



This paper considers reinforcement learning in decentralized, partially observable environments. It presents an "information theoretic framework based on rate distortion theory which facilitates analysis of how well fully decentralized policies are able to reconstruct the optimal solution" and presents an extension "that addresses which nodes an agent should communicate with to improve the performance of its individual policy." The paper presents some proof of concept results on an optimal power flow domain. On the positive side, the idea of applying rate distortion to this problem is interesting and as far as I can tell novel. I rather like it and think it is a promising idea. A major problem, however, is that even though the paper is motivated from the perspective of sequential problems, the formalization is (1) is a static problem. (The authors should describe in how far this corresponds to a DCOP problem). In fact, I believe that the approach is only approximating the right objective for such 1-shot problems: the objective is to approximate the optimal *centralized* policy p(u* | x). However, such a centralized policy will not take into account the affect of its actions on the distribution of information over the agents in the future, and as such much be sub-optimal from in a *decentralized* regime. Concretely: the optimal joint policy for a Dec-POMDP might sacrifice exploiting some information if it can help maintaining predictability and therefore the future ability to successfully coordinate. Another weakness of the proposed approach is in fact this dependence on the optimal centralized policy. Even though computing the solution for a centralized problem is easier from a complexity point of view (e.g., MDPs are in P), this in practices is meaningless since---save very special cases---the state and action spaces of these distributed problems are exponential in the number of agents. As such, it is typically not clear how to obtain this centralized policy. The paper gives a highly inadequate characterization of the related work. For instance just the statement "In practice however, POMDPs are rarely employed due to computational limitations [Nair et al., 2005]." is wrong in so many ways: -the authors mentions 'POMDP' but they cite a paper that introduces the 'networked distributed POMDP (ND-POMDP)' which is not a POMDP at all (but rather a special case of decentralized POMDP (Dec-POMDPs)). -the ND-POMDP makes very strong assumptions on the interactions of the agents ('transition and observation dependence'): essentially the agents cannot influence each other. Clearly this is not the use case that the paper aims at (e.g., in power flow the actions of the agents do affect each other), so really the paper should talk about (perhaps factored) Dec-POMDPs. -Finally, this statement seems a way to try and dismiss "using a Dec-POMDP" on this ground. However, this is all just syntax: as long as the goal is to solve a decentralized decision making problem which is not a very special sub-class** one can refuse to call it a Dec-POMDP, but the complexity will not go away. **e.g., see: Goldman, C.V. and Zilberstein, S., 2004. Decentralized control of cooperative systems: Categorization and complexity analysis. Also the claim "this method is the first of its kind to provide a data-driven way to learn local cooperative policies for multi-agent systems" strikes me as rather outrageous. I am not quite certain what "its kind" means exactly, but there are methods for RL in Dec-POMDPs that are completely decentralized, e.g.: Peshkin L, Kim KE, Meuleau N, Kaelbling LP. Learning to cooperate via policy search. In Proceedings of the Sixteenth conference on Uncertainty in artificial intelligence 2000 Jun 30 (pp. 489-496). Morgan Kaufmann Publishers Inc. An overview of Dec-POMDP RL methods is given in: Oliehoek, F.A. and Amato, C., 2016. A concise introduction to decentralized POMDPs. Springer International Publishing. All in all, I think there is an interesting idea in this submission that could be shaped into good paper. However, there are a number of serious concerns that I think preclude publication at this point.